# A Machine Learning Model to Predict Survival and Therapeutic Responses in Multiple Myeloma

**DOI:** 10.3390/ijms24076683

**Published:** 2023-04-03

**Authors:** Liang Ren, Bei Xu, Jiadai Xu, Jing Li, Jifeng Jiang, Yuhong Ren, Peng Liu

**Affiliations:** 1Department of Hematology, Zhongshan Hospital, Fudan University, Shanghai 200032, China; 2Department of Medical Oncology, Zhongshan Hospital, Fudan University, Shanghai 200032, China; 3Cancer Center, Zhongshan Hospital, Fudan University, Shanghai 200032, China

**Keywords:** multiple myeloma, ubiquitin proteasome pathway, proteasome inhibitors, single-cell RNA-sequencing, machine learning

## Abstract

Multiple myeloma (MM) is a highly heterogeneous hematologic tumor. Ubiquitin proteasome pathways (UPP) play a vital role in its initiation and development. We used cox regression analysis and least absolute shrinkage and selector operation (LASSO) to select ubiquitin proteasome pathway associated genes (UPPGs) correlated with the overall survival (OS) of MM patients in a Gene Expression Omnibus (GEO) dataset, and we formed this into ubiquitin proteasome pathway risk score (UPPRS). The association between clinical outcomes and responses triggered by proteasome inhibitors (PIs) and UPPRS were evaluated. MMRF CoMMpass was used for validation. We applied machine learning algorithms to MM clinical and UPPRS in the whole cohort to make a prognostic nomogram. Single-cell data and vitro experiments were performed to unravel the mechanism and functions of UPPRS. UPPRS consisting of 9 genes showed a strong ability to predict OS in MM patients. Additionally, UPPRS can be used to sort out the patients who would gain more benefits from PIs. A machine learning model incorporating UPPRS and International Staging System (ISS) improved survival prediction in both datasets compared to the revisions of ISS. At the single-cell level, high-risk UPPRS myeloma cells exhibited increased cell adhesion. Targeted UPPGs effectively inhibited myeloma cells in vitro. The UPP genes risk score is a helpful tool for risk stratification in MM patients, particularly those treated with PIs.

## 1. Introduction

Multiple myeloma (MM) is a malignant tumor characterized by abnormal plasma cell clonal expansion in the bone marrow, and is typically accompanied by detectable monoclonal immunoglobulin protein in the peripheral blood or the urine [1]. MM is a heterogeneous disease in a biological context, which is further reflected in the clinical one [2]. Thus, accurate risk stratification is indispensable for MM patients. The International Staging System (ISS) is the most widely used prognostic risk system in clinical practice [3]. However, ISS did not contain individual cytogenetic markers, which are key factors in defining the pathogenesis of myeloma and are strongly related to the prognosis of patients [4,5]. Subsequently, the new prognostic staging systems take into account both ISS and high-risk chromosomal abnormalities, named the Revised International Staging System (R-ISS) and the second Revision of the International Staging System (R2-ISS), were introduced in succession [6,7]. Its efficacy varied in different cohorts [8,9,10,11,12]. As a result, more precise risk scoring systems are needed [13].

For MM patients, abnormal plasma cells mostly produce large quantities of monoclonal immunoglobulin protein (known as M protein), which are misfolded or unfolded [14]. Physiologically, these proteins can trigger a series of downstream signal transduction pathways to be degraded [15,16,17]. These handling pathways were crucial for the survival of myeloma cells [18]. Among these processes, the ubiquitin proteasome pathway (UPP) played a core role, and the strategy that targeted this pathway with proteasome inhibitors (PIs) has achieved promising results [19,20]. Despite that, there is a subset of MM patients who are primarily resistant to PIs and early relapse after first-line therapy, which is strongly associated with inferior survival [21,22]. We must look for new targets to inhibit the ubiquitin proteasome pathway and thus kill myeloma cells.

In this study, we constructed a scoring system that used genes involved in the ubiquitin proteasome pathway to identify MM patients who would benefit more from PIs. Its values were verified in two cohorts with multiple clinical features. A nomogram then combined gene signatures, and ISS was developed to estimate the overall survival (OS) in MM patients. Single-cell transcriptome analysis of 6 MM patients was used to dissect mechanisms of gene signatures for PIs resistance. Finally, the related genes were studied in human myeloma cell lines.

## 2. Results

### 2.1. Subject Selection and Baseline Patient Characteristics

Based on the KEGG pathway gene sets, we selected “KEGG_UBIQUITIN_MEDIATED_PROTEOLYSIS” and “KEGG_PROTEASOME” as related gene sets. A total of 181 genes were included from the gene sets of “c2.cp.kegg.v7.4.symbols” called ubiquitin proteasome pathway associated genes (UPPGs). GSE9782 (N = 264) and CoMMpass (N = 737) datasets were used for subsequent analysis. The basic characteristics were shown in Table 1. In brief, the GSE9782 dataset acted as a training cohort, so the CoMMpass dataset played the character of the validation cohort. Patients in the training cohort were aged 27–86 years (median 61 years), and those in the validation cohorts had a median age of 63 (range: 27–93).

### 2.2. Construction of a Prognostic Gene Signature

To identify UPPGs associated with OS, univariate Cox regression analysis was performed in the training cohort. Twenty genes were significantly related to OS (*p* < 0.001), and six genes were considered protective factors and might prolong the survival of patients (hazard ratio (HR) < 1, Figure 1A). Twenty genes were then subjected to LASSO regression analysis, and 10-fold cross-validation was applied to determine the robust markers for outcomes of MM patients (Figure 1B,C). Eventually, nine genes were confirmed and developed UPPRS. The UPPRS of individual patients was calculated using the following formula: UPPRS = (−0.21852 × expression value of CDC27) + (0.089995 × expression value of CUL1) + (0.189656 × expression value of ELOC) + (−0.01349 × expression value of PML) + (−0.13778 × expression value of UBE2B) + (0.000666 × expression value of UBE2D1) + (0.265798 × expression value of PSMB4) + (0.007926 × expression value of PSMC2) + (0.292179 × expression value of PSME3). We also checked nine genes in the training cohorts by multivariate Cox regression analysis, as shown in Figure 1D, and six genes were independent risk factors for the survival of MM patients.

### 2.3. Evaluation of UPPRS in MM Cohorts

Based on the above formula, the UPPRS of each patient was calculated and patients were divided into high and low UPPRS groups according to the median UPPRS in the training cohort. The relationship between UPPRS and survival status, and the expression of UPPGs in the training cohort, are depicted in Figure 2A. More deaths were found to be associated with higher UPPRS. Among 9 UPPGs, those with HR > 1, such as PSMB4, ELOC, CUL1, PSME3, UBE2D1, and PSMC2, showed higher expression in the high-UPPRS group; as for those HR < 1, UBE2B, CDC27, PML were enriched in the low-UPPRS group. The Kaplan–Meier curve demonstrated that patients in the high UPPRS group had significantly inferior OS compared with the low UPPRS group in the training cohort (Figure 2B). The sensitivity and specificity of the UPPRS were evaluated by ROC analysis. In the training cohort, areas under the curve (AUC) for 1- and 3-year OS rates were 0.71 and 0.92, respectively (Figure 2C).

The predictive performance and robustness of UPPRS were assessed in the MMRF cohort. Patients in the cohort were categorized into a low and a high UPPRS group, and similar results were observed in the validation cohort. The distribution of UPPRS and survival status are visualized in Figure 2D. The survival analyses indicated that OS in the high UPPRS group was significantly worse than patients in the low UPPRS group (Figure 2E). The AUCs for 1-, 3-, and 5-year OS time prediction of UPPRS in the validation cohort were 0.68, 0.66, and 0.64, respectively (Figure 2F). These results demonstrated that UPPRS had an excellent performance in estimating the survival of MM patients.

### 2.4. Clinicopathological Features with UPPRS

Next, we studied the relevance between UPPRS and some traditional clinical parameters. In Figure 3A, age, sex, albumin (ALB), serum β2-microglobulin (β2-MG), ISS, and survival status are summarized in MM patients. We can learn that ISS and β2-MG were significantly related to UPPRS in the training cohort other than survival status. These results were also verified in the validation cohort. However, we also found that there were more female patients in the high UPPRS group in the validation group (*p* < 0.05). We further compared the UPPRS among different survival times. In both cohorts, patients who died during follow-up times had elevated UPPRS, which was more evident in those patients who had short OS (Figure 3B). Univariate cox analyses were performed in both cohorts, and UPPRS were significantly associated with OS (*p* < 0.001, Appendix A). Multivariate cox revealed that UPPRS, β2-MG and ISS were independent prognostic indicators in both cohorts (Figure 3C). Compared to the traditional risk factors, tROC analysis indicated UPPRS was the most robust and accurate predictor for MM patients’ survival outcomes (Figure 3D). Collectively, the above results showed that UPPRS could be a surrogate marker for the tumor burden and risk-stratified factor for MM patients.

### 2.5. UPPRS and Patients’ Response to PIs

To some extent, MM patients’ responses to PIs depended on the ubiquitin proteasome pathway. A total of 169 patients treated with bortezomib (BTZ) had evaluable status in the training cohort. We found that high UPPRS patients tended to have more portion of NC/PD status (NC, no change; PD, progression disease) than that in the low UPPRS group (57.3% vs. 39.7, *p* < 0.05). Further, there were more patients with high UPPRS in the status of NC/PD part (65.5% vs. 48.2%, *p* < 0.05, Figure 4A). In the validation cohort, the number of MM patients whose first-line therapy contained PIs such as bortezomib, or carfilzomib was 646, and the same results were observed (Appendix A). GSEA analysis showed a high UPPRS of the training cohort that was significantly associated with proteasome and protein processing in endoplasmic reticulum pathways, which confirmed that ubiquitin proteasome pathway was overactive in the high UPPRS group (Figure 4B). This feature was verified in the validation cohort (Appendix A). Interestingly, we also found that some classical tumor-related pathways were suppressed in the high UPPRS group such as the JAK-STAT, PI3K-Akt signaling pathway and pathways in cancer. On the other side, citrate cycle oxidative phosphorylation pathways were significantly enriched, as opposed to the low UPPRS group in both cohorts. Patients with evaluable disease status were classified into two different groups according to PI response: those with NC/SD and PD response (SD, stable disease) into a nonresponse (NR) category, and the rest were assigned to a response (R) group [23]. We compared the 9 UPPGs between the two different units, as shown in Figure 5C, with CDC27 considered as a protective value that was downregulated in the NR group (*p* < 0.01); CUL1, on that contrary, notably upregulated in the NR unit (*p* < 0.001); and the others did not show significant statistical differences. However, we further found that four risk factors, PSME3, PSMB4, UBE2D1, and ELOC were upregulated in the NR set in validation cohorts (Appendix A). A bubble heatmap depicted the connection between common agents treating MM and gene expression in Figure 4D. UBE2D1 conferred bortezomib resistance while CUL1, PSMC2, and UBE2B were sensitive to BTZ. CUL1 and UBE2B also exhibited dexamethasone sensitivity. Panobinostat’s response was found to be negatively related to UBE2D1.

### 2.6. A prognostic Model Combined UPPRS and Other Clinical Factors

A total of 918 patients with complete clinical annotations, including age (<65 or ≥65), sex, ISS, and UPPRS (low or high), were applied to build a decision tree to optimize risk stratification in MM patients’ outcomes. As shown in Figure 5A, MM patients were divided into three different divisions based on two powerful factors: UPPRS and ISS. Strangely, we found UPPRS substituted ISS-II/III in the decision tree, and survival curves indicated that three risk subgroups had remarkedly different survival outcomes (*p* < 0.0001, Figure 5B). UPPRS can also pick out the high-risk patients of the ISS-II/III subgroup in the two cohorts (Appendix A). Importantly, UPPRS can further stratify MM patients who were thought to be intermediate risk (ISS/R-ISS = II)––those with high UPPRS had worse OS than patients with low UPPRS in both cohorts (Figure 5C,D and Appendix A).

With the intention of constructing a nomogram to predict OS, two independent risk factors, ISS and UPPRS, were brought into the model (Figure 6A). The nomogram demonstrated larger AUCs compared to ISS and UPPRS in the two cohorts (Appendix A). Correspondingly, the nomogram also showed good prediction ability in estimating the survival time (Appendix A), especially in those patients taking PIs in both units (Figure 6B and Appendix A). The nomogram exhibited the most robust capacity, with the average AUC reaching 0.8 for OS prediction compared to other prognosis-associated risk factors combined with ISS in the validation group (Figure 6C). These results suggested that the nomogram could quite possibly predict the outcomes of MM patients.

### 2.7. UPPRS Analysis in Single-Cell Level

The scRNA sequencing data derived from 3 NDMM and 3 RRMM samples were analyzed (Appendix A). After quality control, 40,675 cells were clustered into 7 cell types (Figure 7A). A total of 11,392 myeloma cells were identified using CopyKAT analysis. We found 1703 genes were significantly different expressions between the NDMM and RRMM samples (Appendix A). Gene Ontology (GO) analysis was then performed on different expressed genes (DEGs), we can find that GO functional groups mainly focus on the proteasome-mediated ubiquitin-dependent protein catabolic process, which highlighted the critical role that the ubiquitin-proteasome system played in the process of PIs resistance (Figure 7B). We check the expression of 9 UPPGs in the formula among two types of samples, among which most show significant differences, except CUL1 and PML. PSMB4 was notably increased in the recurrence samples (Figure 7C). The UPPRS of each cell was achieved using the above formal, low-, and high-risk UPPRS myeloma subtypes that were generated. RRMM samples were found to have the higher UPPRS than the NDMM part (0.14 vs. 0.22, *p* < 0.001), and, correspondingly, high-risk UPPRS cells were more concentrated in the RRMM group (62.9% vs. 43.8%, *p* < 0.001, Figure 7D). Different subtypes of myeloma cells might have distinct interactions with components in the tumor microenvironment. Cell-cell communications are shown in Appendix A. We can tell that there was a similar interaction network between two subtypes of myeloma cells (Figure 7E). To a specific signal pathway, ligand LAMC1 with its receptor CD44 was active from the high-risk UPPRS myeloma cells to the rest of the cell types, and ligand–receptor pair HLA-DMA-CD4 was also found to increase from high-risk UPPRS myeloma cells to monocytes. The enhanced LAMC1-CD44 pathway was exclusively seen in high-risk UPPRS myeloma subtypes to other cells (Figure 7F). The APRIL signaling pathway was active between monocytes and neutrophils (TNFSF13), with its receptor (TNFRSF13B) in high-risk UPPRS myeloma cells (Appendix A).

### 2.8. External Validation in Cell Lines

We first examined gene expression levels in the CCLE database [24]. As shown in Figure 8A, PSMB4 was the most abundant gene, and, on the contrary, was UBE2D1 was the least abundant, which was interpreted by the mutation profile in Figure 8B, with PSMB4 mostly amplified up to 9%, and with deletion more common in UBE2D1 [25]. We then screened potential drugs related to UPPRS by CellMiner, a website to explore gene expression and drugs based on the NCI-60 cell line set [26], and drugs that had been approved by the Food and Drug Administration (FDA) or under clinical trials were included. The nine strongest correlation pairs are shown in Figure 8C. However, the correlations were not significantly high: ranked first, for example, was SCH-900776:PML (cor = 0.496, *p* < 0.001). As a result, we further validated the effects of inhibiting the expression of related genes in human myeloma cell lines using other compounds. Among the nine genes, UBE2B inhibitor TZ9 was accessible, two myeloma cell lines (MM.1S, H929) were treated with a range dose of TZ9 for 48 h, cell viability was significantly reduced and analyzed by CCK8 assay, and H929 was more sensitive to TZ9 with IC50 was 7.757 μM (Figure 8D).

## 3. Discussion

MM is highly heterogeneous in biological and clinical contexts, and a universal risk stratification tool was absent [27]. The ubiquitin proteasome pathway is typically significant in the pathogenesis of MM [28,29]. We constructed a 9 UPP-related gene risk score to stratify MM patients, and it was named UPPRS. UPPRS was further confirmed to be an independent prognostic factor in two separated datasets besides ISS. A nomogram incorporating UPPRS and ISS was generated. The nomogram indicated good sensitivity and specificity in estimating the outcomes of MM patients.

Among the members of the UPPRS formula, most have been reported to be associated with MM. CUL1, a core component of E3 ubiquitin-protein ligase complexes, whose overexpression led to inferior PFS (PFS is progression-free survival) and OS [30]. CUL1 was identified as an independent risk factor with an unfavorable survival outcome in the training cohort. CUL1 was also upregulated in BTZ-resistant patients. Studies showed that the knockdown of CUL1 can sensitize or further overcome myeloma cells’ resistance to PIs [30], highlighting the function of CUL1 in treating MM patients. PSMB4, a subunit of proteasome, also had higher expression in primary refractory MM patients who either failed to respond or experienced early relapse after a bortezomib-containing regimen. It showed PSMB4 was an unfavorable factor in the OS of MM, which corresponds to the above formula [31].

UPPRS, which was derived from the nine genes, showed some clinical benefit. More abnormal β2-MG patients were gathered in the high UPPRS group. Correspondingly, advanced-stage MM patients were more common in the higher part. These suggested that UPPRS could be an indicator of tumor burden in MM patients. In another way, UPPRS can predict the response treated by PIs. Low UPPRS patients may expect better outcomes treated with upfront therapy containing PIs. GSEA results showed increased proteasome activity concentrated in the high UPPRS group, which consist of a larger portion of no-responders to BTZ. These results were supported by a previous report in MM cell lines and other similar research [30,32]. The expressions of certain genes were related to the sensitivity to BTZ.

The nomogram combined ISS and UPPRS exhibited the highest accuracy and discrimination in prediction of MM patients’ survival compared to the two revisions of ISS, including R-ISS and R2-ISS. More than half of MM patients were classified into stage-II, and this group of patients have different risk levels of survival [33]. The UPPRS can discriminate patients into various prognoses, so that the nomogram can make up the defects of the existing models to some extent. Considering the wide use of PIs in the clinical context, the nomogram may have prospects in the routine of treating MM.

The scRNA-seq analysis confirmed the UPP in the access of PIs that are resistant. After applying UPPRS to individual myeloma cells, in our case, low and high-risk UPPRS myeloma cells coexist in NDMM and RRMM samples, and more high-risk UPPRS myeloma cells were found in RRMM samples. We can tell that there were heterogeneous subpopulations existing in MM patients at first, and under the selective pressure of the drug, clone patterns of MM cells have evolved. This highlighted the fact that MM is a highly heterogeneous disease [33,34]. Some immune cells could help myeloma cells escape from death and become drug resistant or else relapse [35]. High-risk UPPRS myeloma cells interact actively with immune cells via the LAMININ pathway, and this may be associated with increased cell-adhesion of myeloma cells and, thus, the acquisition of drug resistance [36].

Finally, nine genes in the formula were reviewed in the CCLE database and cBioPortal. Given that we have not screened any clinical or pre-clinical drugs that are strongly related to UPPGs in CellMiner database, TZ9, a UBE2B-inhibitor, was found to inhibit the growth of MM cell lines in vitro experiment, which provided a new insight regarding the treatment of relapsed MM patients.

While we identified nine potential genes related to OS and therapeutic responses of MM patients, there are some limitations to this study. First, selection bias cannot be ruled out due to the fact that certain public datasets were adopted. Second, although a risk-score formal was developed based on UPPGs, the weight of each gene needs to be testified in large prospective studies. Third, the specific way that UPP changed during the acquisition of drug resistance still needs to be clarified, and further detailed research is needed to explore the underlying mechanism.

Above all, we developed a risk score system on the grounds of UPP genes in MM. Our UPPRS was associated with MM patients’ response to PIs, and could thereby evaluate the prognostics of patients. These values of UPPRS were verified in the single-cell transcriptomic-based dataset. Due to the accessibility of the respective inhibitor, the vitro experiment testified to the importance of related genes for the survival of MM cells. Combined UPPRS and ISS into a nomogram could predict the outcome of MM patients in a better way.

## 4. Materials and Methods

### 4.1. Data Collection

The mRNA data and clinical features were obtained from the GEO database and MMRF CoMMpass study Researcher Gateway (https://www.ncbi.nlm.nih.gov/geo/query/acc.cgi?acc=GSE9782 (accessed on 8 July 2021) and https://research.themmrf.org (accessed on 20 July 2021)). The microarray dataset consisted of 264 MM patients from GSE9782 and was considered as a training cohort, and CoMMpass cohorts included 737 MM patients for validation. Log2 transformation and normalization were performed for the expression data. Groupings of ubiquitin proteasome pathway associated genes (UPPGs) were picked in the gene sets of “c2.cp.kegg.v7.4. symbols” downloaded from the Molecular Signatures Database (MSigDB).

### 4.2. Constructing of the Prognostic Gene Signatures

Univariable Cox regression analysis was applied in the GSE9782 training cohorts to screen UPPGs associated with OS. UPPGs (*p* < 0.001) were then submitted to the least absolute shrinkage and selection operator (LASSO) regression analysis. A prognostic model of UPPGs was established and the ubiquitin proteasome pathway risk score (UPPRS) was generated for individual patients using the following formula: UPPRS = (β1 × expression value of gene 1) + (β2 × expression value of gene 2) + … + (βi × expression value of gene i), and β refers here to the coefficient determined in the LASSO regression analysis. MM patients were divided into high- and low-risk UPPRS groups based on the median UPPRS in the cohorts.

UPPRS was verified by univariate and multivariate Cox regression analysis in the training cohort, and the Kaplan–Meier survival curve was plotted between the two groups. The associations between UPPRS and survival and other clinical characteristics were studied. The accuracy and repeatability of UPPRS in the prediction of OS in MM patients were evaluated by receiver operating characteristic (ROC) curves and calibration curves. The above appraisals were repeated in the validation cohort.

### 4.3. Constructing of the Prognostic Nomogram

A nomogram was developed by combing the clinical parameters and UPPRS using the independent risk factors in the training cohort identified by multivariate Cox regression. ROC analysis and calibration plots were performed to examine the efficacy of the nomogram in both cohorts.

### 4.4. Single-Cell Samples and Data Processing

BM samples from three newly diagnosed multiple myeloma (NDMM) and three Relapse/Refractory multiple myeloma (RRMM) patients were collected at Zhongshan Hospital, Fudan University. The diagnosis definition of MM was adopted from the International Myeloma Working Group (IMWG) criteria of 2014 [37]. RRMM patients are referred as patients relapse after taking PIs-based therapy. Written consent was obtained from each patient. Details of patients are summarized in Appendix A. The Clinical Research Ethics Committee of the Fudan University-affiliated Zhongshan Hospital approved this study.

BM mononuclear cells (BMMC) were processed into single-cell suspensions, and then the 10 × Genomics system was used to perform single-cell RNA-sequencing (scRNA-seq). The scRNA-seq data were processed and analyzed via R package Seurat (v4.3.0) [38]. Cell type annotation was performed using SingleR and classical markers in the literature. R package CopyKAT was used to estimate the genomic copy number profile of cells [39]. The intercellular communications were investigated and visualized by CellChat [40].

### 4.5. Cell Culture and Reagents

Human myeloma cell lines (HMCLs) MM.1S and H929 were purchased from American Type Culture Collection (ATCC). Cells were cultured in RPMI 1640 medium (Gibco) and supplemented with 10% fetal bovine serum in an atmosphere containing 5% CO_2_, supported by a 37 °C incubator. The UBE2B inhibitor (TZ9) [41] was purchased from Med Chem Express (MCE) and dissolved in dimethyl sulfoxide (DMSO) to a stock concentration of 5 mM, and it was stored at −20 °C.

### 4.6. Cell Viability Assay

Cells were seeded into a 96-well plate at 10,000 cells/well, TZ9 with different concentrations was then added for 48h, and then cell counting kit-8 (CCK-8, Biosharp) was used to assess the cell ability according to the manufacturer’s instruction.

### 4.7. Statistical Analysis

The R software version 4.0.5 was used for statistical analyses. The LASSO regression analysis was performed via R package “rms” [42]. Survival curves were plotted by R package “survival” and compared by log-rank test. The R package “timeROC” was used to assess the sensitivity and specificity of the gene signature and nomogram [43]. Gene set enrichment analysis (GSEA) was performed by the R package “clusterprofiler” [44]. A decision tree was built via the “rpart” package [45]. The IC50 of specific drugs and its corresponding mRNA gene expression were taken from the Genomics of Drug Sensitivity in Cancer (GDSC) and Cancer Therapeutics Response Portal (CTRP) [46,47]. The chi-square test or Fisher’s exact test were applied to make the difference for categorical variables. The expression of genes between groups was compared by the Wilcoxon test. A two-sided *p* < 0.05 was regarded as statistically significant. The *p*-value was shown as the format, * *p* < 0.05; ** *p* < 0.01; *** *p* < 0.001 (ns: no significance).

## Figures and Tables

**Figure 1 ijms-24-06683-f001:**
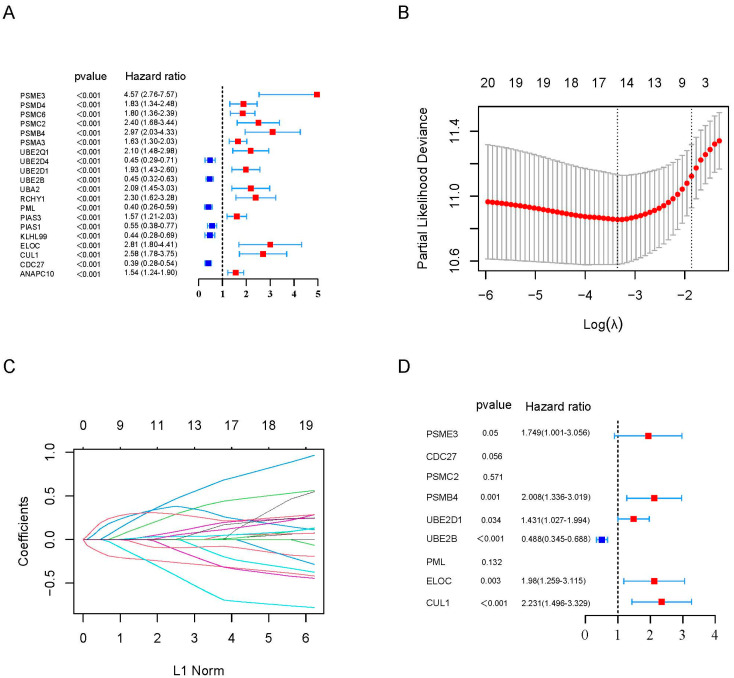
Establishment of the UPPGs–related prognostic signature. (**A**) Univariate cox regression analysis of UPPGs related to MM survival in the training cohort. *p* < 0.001 was considered to be statistically significant. (**B**) LASSO cox regression analysis for the optimal parameter (lambda, λ). (**C**) LASSO coefficient profiles of 20 UPPGs. (**D**) Multivariate cox analysis of nine identified UPPGs associated with MM risk in the training cohort.

**Figure 2 ijms-24-06683-f002:**
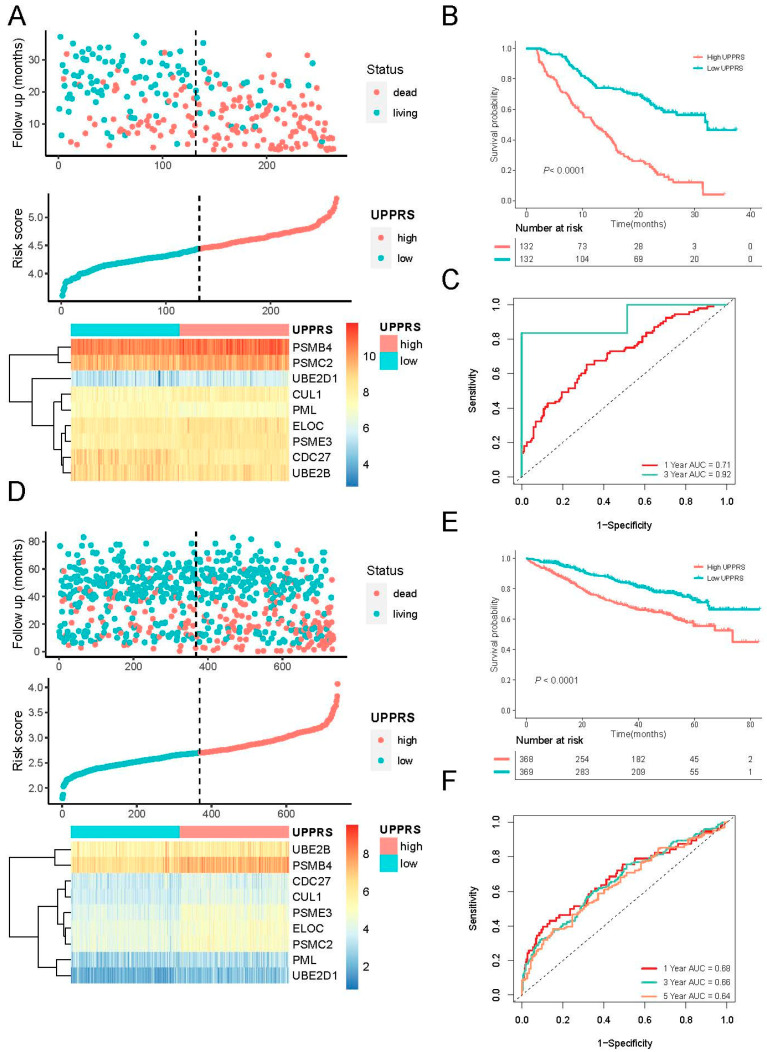
Construction of the UPPRS in the training and validation cohort. (**A**) Survival status, distribution of UPPRS, and expression of the UPPGs in the training cohort. (**B**) Survival curves for different groups in the training cohort. (**C**) Time-dependent ROC analysis of nine-gene signature for 1- and 3-year OS in the training cohort. (**D**) Survival status, distribution of UPPRS, and expression of the UPPGs in the validation cohort. (**E**) Survival curves for different groups in the validation cohort. (**F**) Time-dependent ROC analysis of nine-gene signature for 1-, 3-, and 5-year OS in the validation cohort.

**Figure 3 ijms-24-06683-f003:**
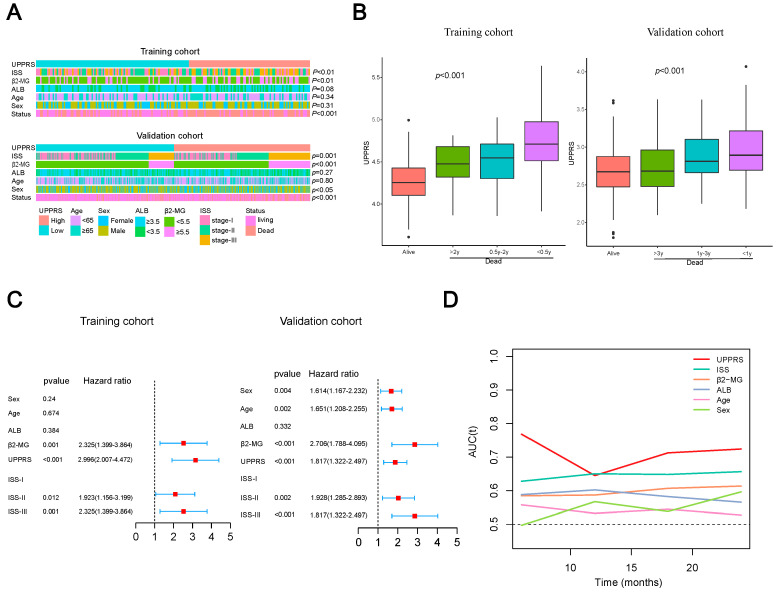
The clinical benefits of UPPRS. (**A**) A heatmap depicting the association between UPPRS and clinicopathological characteristics in two groups. (**B**) UPPRS were significantly elevated in deceased patients, especially in shorter survival groups. (**C**) Multivariate cox analysis of UPPRS combined some common clinical factors in both cohorts. (**D**) tROC analysis of UPPRS and other clinical variables in the prediction of MM patients in the training cohort.

**Figure 4 ijms-24-06683-f004:**
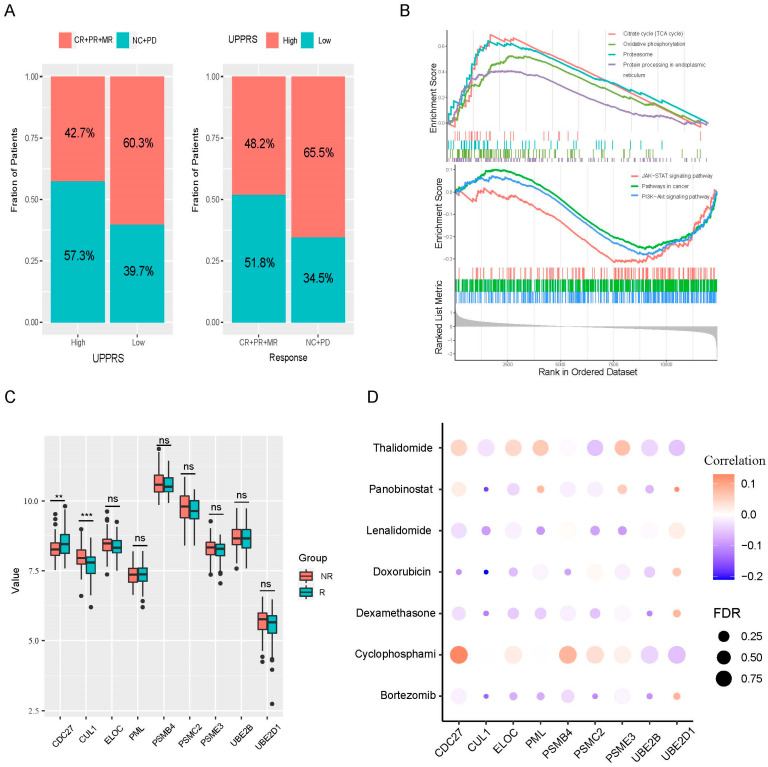
UPPRS can predict the response trigged by PIs. (**A**) The association between UPPRS and clinical response to bortezomib. (**B**) Pathways with significant enrichment in different UPPRS groups in the training group. (**C**) The expression of nine UPPGs in the different response groups in the training cohort. (**D**) The relationships between gene signature expression and common drugs used for MM. PIs, proteasome inhibitors; PD, progression disease; NC, no change; MR, minimal response; PR, partial response; CR, complete response. ** *p* < 0.01; *** *p* < 0.001, ns: no significance.

**Figure 5 ijms-24-06683-f005:**
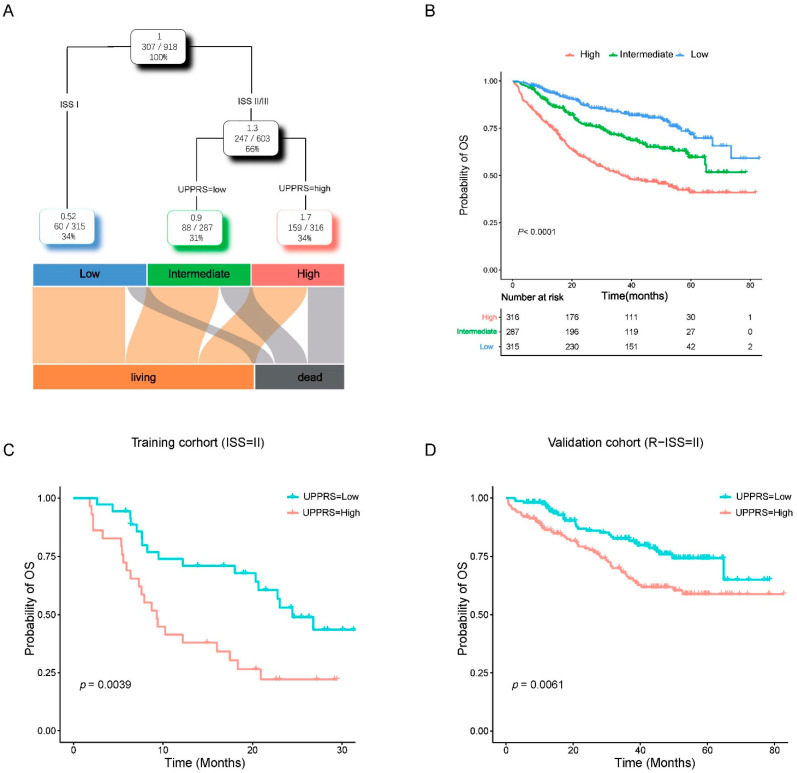
Combination of the UPPRS and ISS improves risk stratification in MM patients. (**A**) A decision tree was developed to stratify MM patients’ risk. (**B**) Performance of decision tree. (**C**) Kaplan–Meier curve for the UPPRS in ISS = II patients in the training cohort. (**D**) Kaplan–Meier curve for the UPPRS in R-ISS = II patients in the validation cohort.

**Figure 6 ijms-24-06683-f006:**
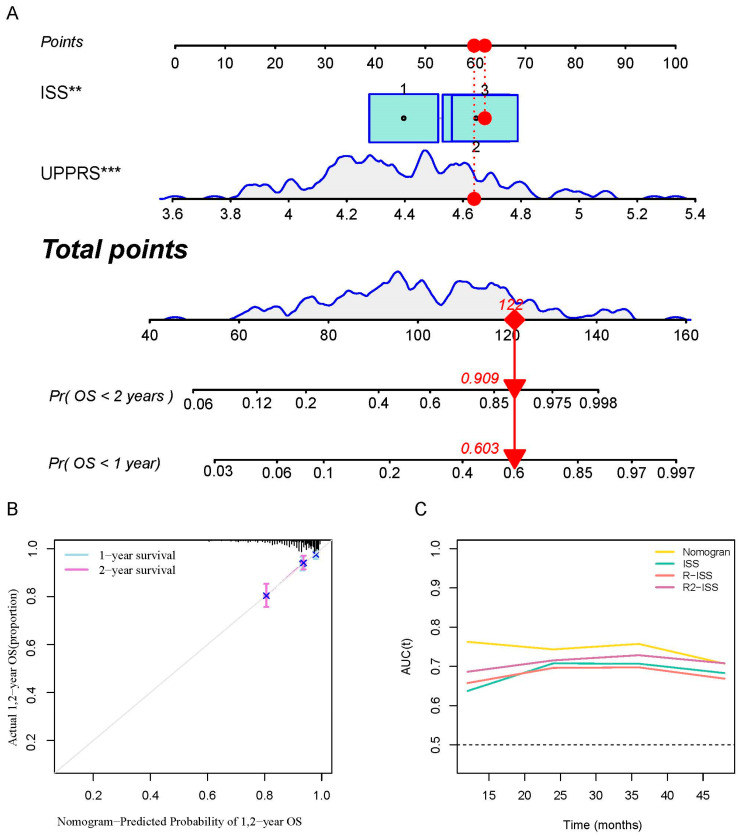
Construction of a nomogram incorporating UPPRS and ISS. (**A**) Details of nomogram. (**B**) Calibration plots of the nomogram at 1- and 2-year OS in the validation cohort that were treated with PIs. (**C**) tROC analysis of nomogram and ISS, R-ISS, R2-ISS in the prediction of MM patients in the validation cohort. ** *p* < 0.01; *** *p* < 0.001.

**Figure 7 ijms-24-06683-f007:**
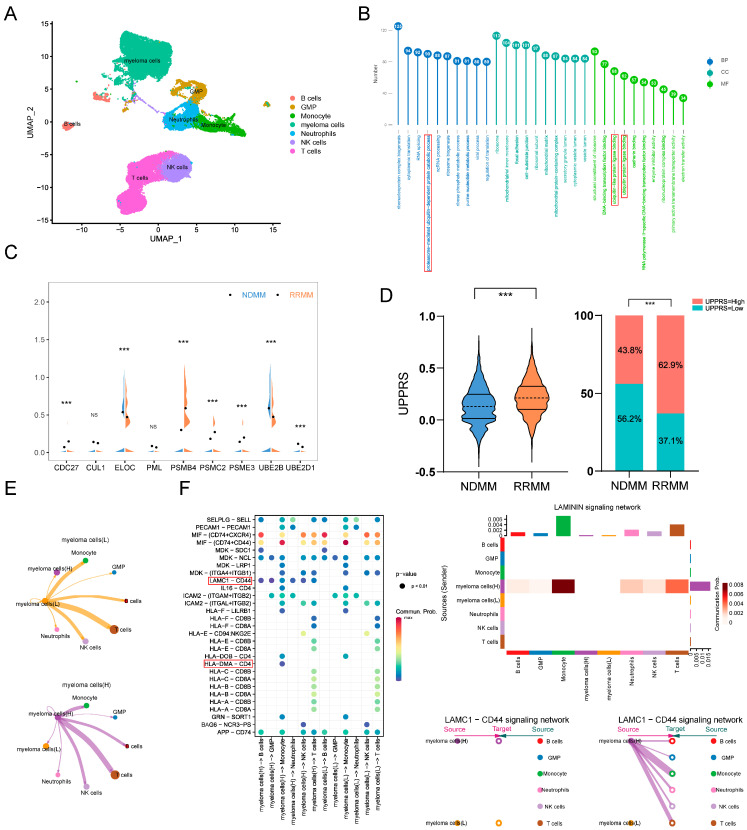
UPPRS analysis at sc−RNA level. (**A**) UMAP plot of 40,675 cells from 3 NDMM and 3RRMM samples. (**B**) GO enrichment analyses on the DEGs between NDMM and RRMM myeloma cells. (**C**) The comparison of 9 UPPGs in UPPRS between NDMM and RRMM myeloma cells. (**D**) The UPPRS in NDMM and RRMM myeloma cells. (**E**) Strength of the cell−cell communications among high−UPPRS, low−UPPRS myeloma cells, and other immune cells. (**F**) Ligand−receptor in the communication of two subtypes of myeloma cells to immune cells. Myeloma cells (H)––high-UPPRS myeloma cells; Myeloma cells (L)––low-UPPRS myeloma cells. *** *p* < 0.001. ns: no significance.

**Figure 8 ijms-24-06683-f008:**
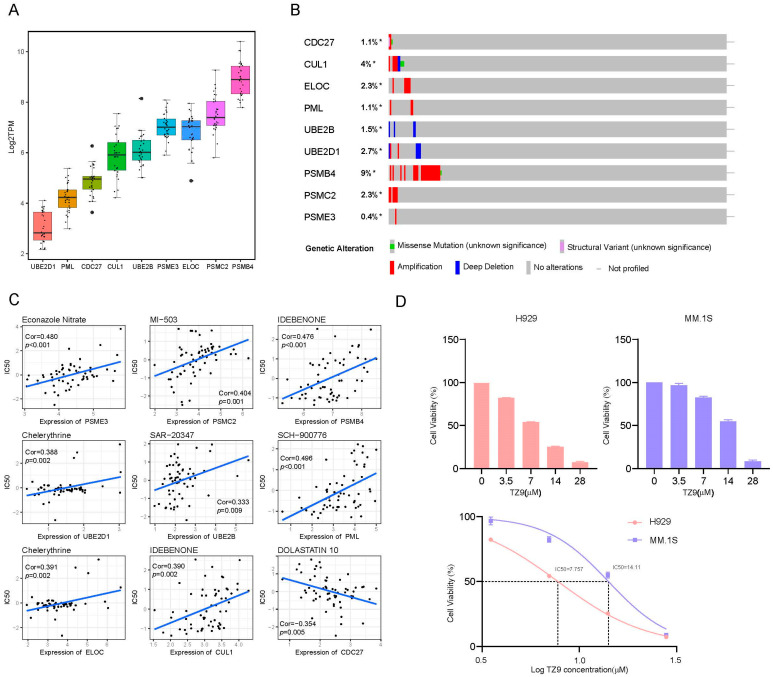
External validation in online databases. (**A**) The expression of UPPGs in HMCLs demonstrated in CCLE. (**B**) The mutation profile of MM samples from cBioPortal. (**C**) Correlation analysis between nine UPPGs and their potential targeted drugs. (**D**) TZ9 treated H929, MM.1S cell lines. HMCLs––Human myeloma cell lines. * *p* < 0.05.

**Table 1 ijms-24-06683-t001:** Baseline clinical features of patients in the training and validation cohort.

Characteristics	Training Cohort GSE9782	Validation Cohort CoMMpass
	N = 264	N = 737
Age, years(median, range)	61 (27–86)	63(27–93)
<65	177 (67%)	407 (55%)
≥65	87 (33%)	330 (45%)
Sex		
Male	159 (60%)	438 (59%)
Female	105 (40%)	299 (41%)
Hemoglobin (g/L)		
≥100	-	454 (62%)
<100	-	283 (38%)
ALB (g/L)		
≥35	167 (63%)	440 (60%)
<35	83 (31%)	297 (40%)
β2-Mg (mg/L)		
<5.5	136 (52%)	503 (68%)
≥5.5	68 (26%)	169 (23%)
LDH (U/L)		
<250	-	496 (67%)
≥250	-	119 (16%)
ISS		
stage-I	69(26%)	246 (33%)
stage-II	65(25%)	261 (35%)
stage-III	68(26%)	209 (28%)
Cytogenetics		
del (17p)	-	70 (9.5%)
t (4; 14)	-	69 (9.4%)
t (4; 16)	-	23 (3.1%)

ALB, albumin; β2-MG, serum β2-microglobulin; LDH, lactate dehydrogenase; ISS, International Staging System.

## Data Availability

The data used to support the findings of this study are available either online or from the corresponding author upon request.

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
