# Peer review of "A Machine Learning Model to Predict Survival and Therapeutic Responses in Multiple Myeloma"

_ijms, 2023, doi:10.3390/ijms24076683_

Round 1
Reviewer 1 Report
The authors presented the studies about a scoring system to identify MM patients benefit more from PIs, with multiple clinical features and e single-cell transcriptomic based dataset. Overall it seems that this paper is very well written and the logic behind each step and conclusion is well supported and suggests the possibility for future study. As such, I recommend the present work for publication, after the authors have addressed the following minor issues:
-
In Table 1, the training cohort and validation cohort have huge differences in distribution among many axes. It would be very helpful if the authors can justify the impact of this distribution gao to the models;
-
Overall, it’s seems to be small dataset with limited features, the author needs to justify in this scenario why we choose machine learning but not traditional model (linear regression). Some comparison would be very helpful.
Author Response
Reviewer 1. Comments and Suggestions for Authors
The authors presented the studies about a scoring system to identify MM patients benefit more from PIs, with multiple clinical features and e single-cell transcriptomic based dataset. Overall it seems that this paper is very well written and the logic behind each step and conclusion is well supported and suggests the possibility for future study. As such, I recommend the present work for publication, after the authors have addressed the following minor issues:
1.
In Table 1, the training cohort and validation cohort have huge differences in distribution among many axes. It would be very helpful if the authors can justify the impact of this distribution gao to the models;
2.
Overall, it’s seems to be small dataset with limited features, the author needs to justify in this scenario why we choose machine learning but not traditional model (linear regression). Some comparison would be very helpful.
Response 1: Thanks for your suggestion. We conducted a comprehensive search of publicly available datasets on the internet, utilizing inclusion criteria centered around transcriptome sequencing databases that contained pertinent PI-based regimens and survival information, eventually, the two datasets were included. We compared the distributions of variables between the two datasets listed in Table 1. In our analysis, it was observed that the gender distribution was comparable between the two datasets (p=0.82), whereas the distributions of other variables, age, ALB, β2-MG, and ISS, exhibited significant differences between the two datasets(p<0.001). Despite the examination of several variables in this study, it is important to note that the variables analyzed were limited to UPPGs, and as a result, the predicted indicator generated (UPPRS) does not demonstrate any correlation with the variables presented in Table 1. Finally, the machine learning models generated in this study demonstrate robust predictive capabilities in both datasets, whether in predicting survival outcomes or treatment responses. Therefore, the variable distribution presented in Table 1 does not impact the utility of the models.
Response 2: Thanks for your suggestion. The multivariate Cox regression analysis is the most commonly used model for survival analysis. When utilizing the Cox proportional hazards regression model, adequate sample size is a critical consideration. Specifically, a minimum number of positive outcome events, equivalent to 10-15 times the number of predictor variables, is required to ensure statistical power and reliable model performance. In the training cohort, the number of positive outcome events is 157. However, there are 20 variables (p<0.001) that require additional screening. So the data currently available is not appropriate for conducting multivariate Cox regression analysis. We then adopted machine learning analysis methods to build predictive models. Nevertheless, We still used the multivariate Cox regression analysis to construct a predictive index. The 7 independent risk factors were left to be incorporated in the new risk-system. The new generated model with C-index of 0.72 in predictive accuracy for OS in the training cohort, made no significant difference compared with UPPRS used in our paper (0.72 vs 0.72, p=0.4) , the same is for AUCs in the training cohort (Figure 1A). However, The AUCs for 1-, 3-, 5-year OS time prediction of new risk-system in the validation cohort were inferior to be predicted by UPPRS( Figure 1B). Therefore, we believe that the existing machine learning models are greater to the traditional model.

Figure 1. Time-dependent ROC analysis of nine-gene signature for 1-,3-year OS in the training cohort(A). Time-dependent ROC analysis of nine-gene signature for 1-,3-,5-year OS in the validation cohort(B).
Reviewer 2 Report
The manuscript with the title of "A machine learning model to predict survival and therapeutic 2 responses in multiple myeloma" was well written and presented the topic appropriately. my only comment is regarding the figures. there are too many graphs and most of them are unreadable due to small font size. I would keep the summary figures in the manuscript and move the detailed ones to an appendix; alternatively, arrange the figures in a way that they are readable and are covering the data.
Author Response
Reviewer 2. Comments and Suggestions for Authors
The manuscript with the title of "A machine learning model to predict survival and therapeutic 2 responses in multiple myeloma" was well written and presented the topic appropriately. my only comment is regarding the figures. there are too many graphs and most of them are unreadable due to small font size. I would keep the summary figures in the manuscript and move the detailed ones to an appendix; alternatively, arrange the figures in a way that they are readable and are covering the data.
Response : Thanks for your suggestion. We appreciate your suggestion regarding the figures and agree that they play a critical role in presenting the research findings effectively. We apologize for the inconvenience caused due to the small font size of the figures, which we acknowledge is partly due to the image compression that occurred during their insertion. In response to your comment, we have made revisions to the manuscript to improve the readability of the figures. We have carefully evaluated all the figures and have decided to remove some of the less relevant ones to avoid overwhelming the reader with too much data.
We have also reorganized the remaining figures in a way that they are more easily readable and are covering the data. Additionally, we have included a summary figure in the manuscript that presents a condensed version of the most critical data. We have also moved the detailed figures to an appendix, as you have suggested, to make them more accessible to readers who are interested in exploring the data in-depth. We believe that these changes will significantly improve the quality of our manuscript and make it more accessible to a broader audience.
Thank you once again for your feedback, which has helped us to improve the quality of our manuscript.